# Risk of Liver Fibrosis Is Associated with More Severe Strokes, Increased Complications with Thrombolysis, and Mortality

**DOI:** 10.3390/jcm12010356

**Published:** 2023-01-02

**Authors:** Emma M. S. Toh, Priscilla Roshini Joseph Ravi, Chua Ming, Amanda Y. L. Lim, Ching-Hui Sia, Bernard P. L. Chan, Vijay K. Sharma, Cheng Han Ng, Eunice X. X. Tan, Leonard L. L. Yeo, Daniel Q. Huang, Mark D. Muthiah, Benjamin Y. Q. Tan

**Affiliations:** 1Yong Loo Lin School of Medicine, National University of Singapore, Singapore 119077, Singapore; 2Division of Endocrinology, Department of Medicine, National University Hospital, Singapore 119074, Singapore; 3Division of Cardiology, Department of Medicine, National University Heart Centre Singapore, Singapore 119074, Singapore; 4Division of Neurology, Department of Medicine, National University Hospital, Singapore 119074, Singapore; 5Division of Gastroenterology, Department of Medicine, National University Hospital, Singapore 119074, Singapore

**Keywords:** FIB-4, liver fibrosis, symptomatic intracranial haemorrhage, ischemic stroke, thrombolysis

## Abstract

The Fibrosis (FIB)-4 index is an established non-invasive test to detect liver fibrosis. Liver fibrosis is postulated to be one of the predictors of the risk of symptomatic Intracranial Haemorrhage (SICH) after intravenous tissue plasminogen activator (IV tPA) therapy, the mainstay of treatment following acute ischemic stroke (AIS). However, SICH is a feared complication of thrombolytic therapy. We aimed to evaluate the association of FIB-4 with outcomes of AIS after IV tPA. Consecutive AIS patients receiving IV tPA from 2006 to 2018 at a single stroke centre were studied in a retrospective cohort study. Multivariable adjusted logistic regression was performed to assess associations of FIB-4 with outcomes. The primary outcome was SICH, and secondary outcomes included functional independence (mRS of 0–2) and mortality measured at 90 days. Among 887 patients (median age: 67 (IQR: 57–77)), 342 had FIB-4 < 1.3 and 161 had FIB-4 > 2.67. A greater proportion of moderate to severe strokes (NIHSS ≥10) occurred in the FIB-4 > 2.67 group (*n* = 142, 88.8%) compared to the FIB-4 < 1.3 group (*n* = 208, 61.2%). Amongst the different stroke subtypes, median FIB-4 was highest in cardioembolic stroke (CES) compared to the 3 other non-CES stroke subtypes (1.90 (IQR: 1.41–2.69)). Following IV tPA, having FIB-4 > 2.67 was associated with an increased rate of SICH (adjusted OR: 4.09, 95% CI: 1.04–16.16, *p* = 0.045) and increased mortality (adjusted OR 3.05, 95% CI: 1.28–7.26, *p* = 0.012). Advanced liver fibrosis was associated with an increased rate of SICH and increased 90-day mortality after IV tPA. The FIB-4 score may be useful for prognostication after IV tPA.

## 1. Introduction

With an ageing population and rise in stroke-attributable risk factors, the prevalence of acute ischemic stroke (AIS) continues to increase, remaining the second leading cause of death worldwide and second most common cause of disability-adjusted life years [1]. Similarly, chronic liver disease is a major cause of morbidity and mortality worldwide. Non-alcoholic fatty liver disease (NAFLD), one of the causes of chronic liver disease, affects one quarter of the world’s adult population and has a rapidly rising global burden, with an estimated 178% increase in NAFLD-related deaths in the United States alone [2].

NAFLD encompasses a spectrum of conditions from the more benign non-alcoholic fatty liver (NAFL) to the more aggressive non-alcoholic steatohepatitis (NASH) [3]. Advanced fibrosis in the setting of NAFLD is associated with increased mortality in liver-related and cardiovascular diseases [4,5]. Notably, AIS and NAFLD share many common risk factors such as hypertension and type 2 diabetes mellitus [3,6], suggesting that the pathophysiology of liver dysfunction and AIS could be related.

AIS patients have been found to have significantly higher levels of liver dysfunction, specifically liver fibrosis [7]. However, there is limited data regarding how fibrosis can affect clinical outcomes in patients after AIS. Moreover, there is a lack of literature evaluating liver fibrosis among AIS patients treated with intravenous tissue plasminogen activator (IV tPA) therapy, the mainstay of treatment for AIS [8]. As the number of stroke patients with NAFLD continues to rise, it is important to identify predictors of post-stroke outcomes, including the feared complication of symptomatic intracranial haemorrhage (SICH) from IV tPA [9]. 

In this study, we characterized liver fibrosis using the Fibrosis (FIB)-4 index, an established non-invasive test to detect liver fibrosis. The FIB-4 index comprises of age, aspartate aminotransferase (AST), alanine aminotransferase (ALT) and platelet count. The aim of this study was to examine the association between FIB-4 and the outcomes of AIS patients after IV tPA therapy.

## 2. Materials and Methods

### 2.1. Data Collection

Our study included consecutive AIS patients who received IV tPA from September 2006 to June 2018 at a comprehensive stroke centre. Patients were assessed by a stroke neurologist to be eligible to receive intravenous thrombolysis according to institutional protocol and American Heart Association/American Stroke Association guidelines at a standard dose of 0.9 mg/kg body weight [8]. All thrombolysed stroke patients underwent standard non-contrast head computed tomography (CT) and brain and neck CT angiography. Other baseline demographics, clinical parameters and ischemic stroke characteristics were collected and tabulated within 24 h of admission for AIS. Diabetes mellitus was defined as pre-existing diagnosis of diabetes mellitus or an HbA1c greater than or equal to 6.5% [10]. The presence of Large Vessel Occlusion (LVO) was defined as occlusions of the first and second segment of the middle cerebral artery (MCA, M1 and M2), the Internal Carotid Artery (ICA) and as well as its terminus, tandem occlusions involving ICA-MCA, or occlusion of the basilar artery. The severity of stroke at presentation was assessed using the National Institute of Health Stroke Scale (NIHSS). Moderate to severe stroke was defined as NIHSS ≥ 10 [11]. This assessment was performed by credentialed nurses who were part of the acute stroke response team. The Trial of Org 10172 in Acute Stroke Treatment (TOAST) criteria was used by the treating stroke neurologist to classify stroke subtypes [12]. Ethics approval was obtained from the National Healthcare Group Domain Specific Review Board (NHG DSRB 2010\00509).

### 2.2. Definitions

The risk of advanced liver fibrosis (≥F3) was assessed using FIB-4, and calculated using the following formula: $\frac{Age\;\left(years\right)\times AST\;\left(U/L\right)}{Platelet\;\left(10^{9}/L\right)\times \sqrt{ALT\;\left(U/L\right)}}$ [13]. The well-established FIB-4 cut-offs of <1.3 and >2.67 were used in this study, with high risk of advanced liver fibrosis (>F3) defined as FIB-4 > 2.67 [14].

### 2.3. Outcome Measured

The primary outcome was symptomatic intracerebral haemorrhage (SICH) after IV tPA therapy for AIS. Secondary outcomes measured included functional independence (mRS of 0–2) and mortality at 90 days. SICH was classified based on the European Cooperative Acute Stroke Study (ECASS) II definition: any type of intracerebral haemorrhage on any post-treatment imaging after the start of thrombolysis and increase of ≥4 NIHSS points from baseline, or from the lowest value within 7 days, or leading to death [15]. The 90-day mRS was evaluated during the follow-up visit to the stroke clinic, and if patients were not able to attend the follow-up visit physically, mRS was evaluated via telephone call instead. 

### 2.4. Statistical Analysis

Statistical tests were performed with R version 4.1.3 [16], and SPSS version 25 [17], and a *p*-value of <0.05 was considered statistically significant. Statistical analysis using Mann–Whitney U test was performed for non-normally distributed continuous variables, and Chi-square analysis for categorical variables. Kruskal–Wallis test was used to determine if there was a statistically significant difference between any of the three FIB-4 groups. Median FIB-4 scores were also evaluated across TOAST subtypes via Kruskal–Wallis test. A binomial logistic regression model was then performed to identify independent associations of fibrosis status with SICH, 90-day functional outcome and mortality. An adjusted multivariate logistic regression model was used to assess the associations between lipid parameters and outcomes, presented as adjusted odds ratios (adj OR) with 95% confidence intervals (CI). Multiple Imputation by Chained Equations (MICE) in SPSS was employed for missing baseline data body mass index (BMI) (14.2% missing), to conduct multivariate regression analyses. The multiple imputation method used was linear regression for scale variables. 30 imputed datasets were created and estimates from the multivariate analysis were pooled using Rubin’s rules [18]. 

## 3. Results

### 3.1. Baseline Characteristics

A total of 887 AIS patients were included in this study. The median age was 67 (Interquartile Range (IQR): 57–77), 535 patients (68.1%) were Chinese and 528 (59.5%) were males. The median presenting NIHSS was 15.5 (IQR: 8–21), where 619 (70.3%) patients suffered moderate or severe strokes. The median FIB-4 in the cohort was 1.53 (IQR: 1.05–2.27). According to stroke subtype, 291 (33.7%) strokes were due to large-artery atherosclerosis, 218 (25.2%) were cardioembolic, 137 (15.9%) were small-vessel occlusion, 11 (1.3%) were strokes of other determined etiology and 207 (24.0%) were strokes of undetermined etiology. Further baseline characteristics are outlined in Table 1. After IV tPA, 47 patients (5.3%) suffered a SICH. At 90 days, 433 patients (48.8%) achieved functional independence (mRS 0–2) and there were 125 deaths (14.1%) (Appendix A).

Of 887 patients, there were 342 (38.6%) patients with FIB-4 < 1.3, 384 (43.3%) patients with 1.3 ≤ FIB-4 ≤ 2.67, and 161 (18.2%) patients with FIB-4 > 2.67. The FIB-4 > 2.67 group was more likely to have atrial fibrillation (31.1% vs. 10.2%), hypertension (75.2% vs. 56.7%), hyperlipidemia (50.3% vs. 47.4%) compared to the FIB-4 < 1.3 group (Table 1). Post hoc analysis showed a statistically significant difference in prevalence of atrial fibrillation (*p* < 0.001) and large vessel occlusion (*p* < 0.001) between the FIB-4 < 1.3 and FIB-4 > 2.67 group (Appendix A).

### 3.2. Stroke Characteristics of Patients with FIB > 2.67

We also analysed the relationship of FIB-4 with several stroke characteristics including stroke severity and TOAST classification. There were more moderate to severe strokes in the FIB-4 > 2.67 group as compared to the FIB-4 < 1.3 group (88.8% vs. 61.2%), and median admitting NIHSS was raised in the FIB-4 > 2.67 compared to FIB-4 < 1.3 group (21 (IQR: 16–24) vs. 12 (IQR: 7–19)), which was confirmed on post hoc analysis (Appendix A). Furthermore, the median FIB-4 was highest in patients with cardiombolic stroke at 1.90 (IQR: 1.41–2.69) compared to other stroke etiology subtypes according to the TOAST classification. This was followed by a median FIB-4 of 1.51 (IQR 1.04–2.44) in patients with stroke of undetermined etiology, compared to 1.42 (IQR 1.02–2.11) in patients with large-artery atherosclerosis and 1.25 (IQR 0.88–1.74) in patients with small-vessel occlusion (all *p* < 0.001). (Figure 1) There was a significantly greater prevalence of cardioembolic stroke subtype and lower prevalence of small-vessel occlusion subtype in the FIB > 2.67 compared to FIB < 1.3 group (Table 1 and Appendix A).

### 3.3. Association of Risk of Advanced Fibrosis and Clinical Outcomes

Among patients with FIB-4 > 2.67 compared to FIB-4 < 1.3, there was a higher risk of SICH (10.0% vs. 2.9%), mortality (30.4% vs. 6.1%) and reduced functional independence (26.7% vs. 62.3%) (Table 2).

The median FIB-4 was higher in patients who developed SICH versus those without (1.96 (IQR: 1.47–3.16) vs. 1.50 (IQR: 1.04–2.25), *p* = 0.001) (Appendix A). After adjustment for age, admitting NIHSS, BMI, gender, hypertension, hyperlipidaemia, diabetes mellitus and atrial fibrillation, presence of large vessel occlusion as well as TOAST subtype, the FIB-4 > 2.67 group was associated with SICH (adjusted OR: 4.089, 95% CI: 1.035–16.160, *p* = 0.045) (Table 2).

A positive association was similarly observed between FIB-4 > 2.67 and mortality. After adjustment, FIB-4 > 2.67 was also associated with mortality (adjusted OR 3.052, 95% CI: 1.283–7.262, *p* = 0.012) (Table 2). However, having FIB-4 > 2.67 was not independently associated with functional independence after IV tPA after adjustment (adjusted OR: 0.998, 95% CI: 0.531–1.876, *p* = 0.995) (Table 2). 

### 3.4. Receiving Operating Characteristic (ROC) Curve Evaluating Predicting Value of FIB-4 Score

ROC curve analysis was additionally employed to further characterise and evaluate the association between the absolute FIB-4 index and the primary outcomes in SICH and mortality. The AUC of using the absolute FIB-4 score was 0.772 (95% De Long Confidence Interval: 0.713–0.831) (Appendix A) for predicting mortality and 0.690 (95% De Long Confidence Interval: 0.573–0.806) (Appendix A) for predicting SICH, indicating good predictive value in patients receiving intravenous thrombolysis after stroke. 

## 4. Discussion

In this study, we evaluated the association between FIB-4 and clinical outcomes among patients treated with IV tPA for acute ischemic stroke (AIS). We observed that patients with FIB-4 > 2.67 had more moderate to severe strokes, were associated with an increased risk of SICH after thrombolysis and increased mortality after acute ischemic stroke and IV tPA compared to patients with FIB-4 < 1.3. Patients with FIB-4 > 2.67 also had higher risk of cardioembolic stroke subtype, which could explain associations with severe stroke. Furthermore, ROC analysis showed the good sensitivity and predictive value of the absolute FIB-4 score for predicting mortality and SICH, that performed better than previously reported AUCs of FIB-4 in a cohort of AIS patients that were not exclusive to thrombolysis [19,20].

Liver biopsy is the gold standard in assessing liver fibrosis, but is limited by its invasive nature, complications and sampling variability [21]. Among readily available non-invasive tests such as AST/ALT ratio, AST to platelet ratio index (APRI), NAFLD fibrosis score (NFS), the FIB-4 index has the best performance, with an AUROC of 0.62–0.80 for the prediction of advanced liver fibrosis (≥F3) [22]. Recently, FIB-4 has been used to predict prognosis after the first episode of ischemic stroke [20] and to predict risk of SICH and other outcomes after mechanical thrombectomy [23]. 

### 4.1. Associations with Cardioembolic Stroke and Moderate and Severe Stroke

The median FIB-4 was highest in patients with cardioembolic stroke as compared to the other stroke subtypes under the TOAST classification. The most important cause of cardioembolic stroke is atrial fibrillation [24]. NAFLD, a common cause of liver fibrosis, is associated with left ventricular diastolic dysfunction [25], which can result in diastolic heart failure—postulated to be an independent predictor of atrial fibrillation [26]. This may contribute to the increased risk of atrial fibrillation in patients at risk of advanced liver fibrosis, and hence increase the risk of cardioembolic stroke, but further studies are required to validate this hypothesis. Secondly, inflammation is a key underlying mechanism regulating liver fibrogenesis [27], and proinflammatory cytokines have a negative inotropic effect, reducing ventricular contractility [28]. Inflammatory damage to endothelial cells (endothelial dysfunction) promotes formation of microthrombi [29] and this coupled with the blood stasis in the ventricles from reduced ventricular contractility increases the risk of cardiac thrombus formation, leading to cardioembolic stroke [24].

Cardioembolic stroke is associated with more moderate to severe strokes. Cardioembolic stroke patients have been reported to have more severe clinical deficits on admission, worse recovery and increased length of stay, compared to other ischemic stroke subtypes [30]. Since cardioembolism is a common cause of acute ischemic stroke [31], this could provide the explanation for our findings that higher FIB-4 is associated with moderate to severe acute ischemic stroke. 

### 4.2. Association with Symptomatic Intracranial Haemorrhage

Symptomatic intracranial haemorrhage (SICH) is one of dangerous complications post IV tPA in AIS, with an incidence of 2% to 7% and outcomes can approach 50% mortality [9]. Studies have found that liver diseases significantly increases plasma tPA levels [32]. tPA is cleared by the mannose receptor on liver endothelial cells and Kupffer cells and the LDL Receptor Protein receptor on parenchymal cells, and our study found that FIB-4 is associated with lower LDL levels. Lower LDL levels could reduce the clearance of tPA, increasing plasma tPA levels, and potentially increase SICH rate [33]. Another hypothesised reason for the increased risk of SICH is endothelial dysfunction. Fibrosis decreases nitric oxide derived from endothelial nitric oxide synthase (eNOS), resulting in endothelial dysfunction and increased SICH [34]. eNOS-derived nitric oxide is also shown to play neuroprotective roles in acute ischemic stroke, such as the modulation of cerebral microvascular tone and preservation of the blood-brain-barrier [35]. Reduced eNOS-derived nitric oxide levels in fibrotic patients could reduce nitric oxide in the brain, reducing neuroprotection in acute ischemic stroke. Moreover, reactive oxygen species and oxidative stress involved in NAFLD and fibrosis progression, which themselves contribute to endothelial dysfunction, could also interrupt microvascular integrity that predispose patients to haemorrhagic transformation [36,37].

### 4.3. Association with Mortality and Functional Outcomes

We found increased rate of mortality in patients with FIB > 2.67, and it is possible that this observation could be related to symptomatic intracranial haemorrhage. However, a majority of our patients did not have SICH, hence alternative mechanisms could explain this. The Framingham Heart study showed that NAFLD, one of the most leading causes of liver fibrosis, was associated with the presence of subclinical markers of atherosclerosis, such as calcium deposits in the coronary arteries, resulting in increased cardiovascular events and mortality [38]. Furthermore, there are many common co-morbidities between liver fibrosis and stroke, including metabolic syndrome which has been postulated to be an independent risk factor for mortality in stroke patients [39]. Finally, worsening liver fibrosis can lead to cirrhosis-associated immune dysfunction. Systemic inflammation can lead to not only impaired circulatory function but also organ failure by a direct effect of the inflammatory mediators on microvascular integrity, cell function and death mechanisms [40].

Although FIB-4 was significantly associated with functional outcome on univariate analysis, the association was not significant on multivariate analysis. The significance of the association returned if admitting NIHSS was not included in the multivariate analysis, suggesting that fibrosis was not an independent predictor of functional outcome and any difference seen on univariate analysis could be driven by an increase in moderate to severe stroke among those with worsened liver fibrosis. 

### 4.4. Strengths and Limitations

Our study is the one of the first to analyse associations of FIB-4 in a large cohort of AIS patients receiving thrombolysis. While the FIB-4 has previously been criticised for possibly inaccurate correlations due to the incorporation of age into the index, our study corrected for age in our multivariate analysis, with no collinearity detected between age and FIB-4. Importantly, our study also accounted for other confounders including BMI, presence of hyperlipidemia which are parameters that are related to NAFLD and possibly fibrosis, and atrial fibrillation which is commonly associated with cardioembolic stroke and therefore fibrosis. Moreover, our study was the first to look at the risk of symptomatic intracranial haemorrhage in acute ischemic stroke patients after IV tPA therapy. Previous similar studies either excluded patients receiving IV tPA [19], or looked at the population of patients receiving mechanical thrombectomy [23].

Our study also has several limitations. The FIB-4 index is a good marker for the risk of fibrosis. Furthermore, this analysis would ideally have benefited from a propensity score matching analysis to increase confidence the effect of fibrosis between matched cohorts, the sample size of the matched cohort limited our ability for a good match to generate accurate conclusions. However, collecting the controlled attenuation parameter (CAP) scores and liver stiffness indices from Fibroscan of the patients could aid in comparison of these indices to FIB-4, and analyse their utility in predicting stroke outcomes and explore their relationships with cardioembolic stroke and AIS severity. It would have been ideal to obtain confirmatory diagnosis of via liver biopsy and utilise the Meta-Analysis of histological data in Viral hepatitis (METAVIR) scoring system for better characterising the stages and extent of fibrosis, but this was limited by the highly invasive nature of a liber biopsy. We additionally did not have information on certain parameters such as Gama-Glutamyl Transferase (GGT) and C-Reactive Protein, which limited our ability to categorize the disease. As this was a single centre retrospective study, causality could not be established, and generalisability to other cohorts may be limited. Further work by increasing sample size or employing other methods of study design and research analysis could be explored to further strengthen the conclusions.

## 5. Conclusions

FIB-4 > 2.67 in stroke patients post IV tPA therapy was associated with increased symptomatic intracranial haemorrhage and mortality. FIB-4 > 2.67 was also associated with higher rates of CES and more moderate to severe AIS. FIB-4 is a novel marker that could be used to prognosticate the outcomes of treating acute ischemic stroke patients with IV tPA, but our data requires further validation in prospective, multicentre studies. 

## Figures and Tables

**Figure 1 jcm-12-00356-f001:**
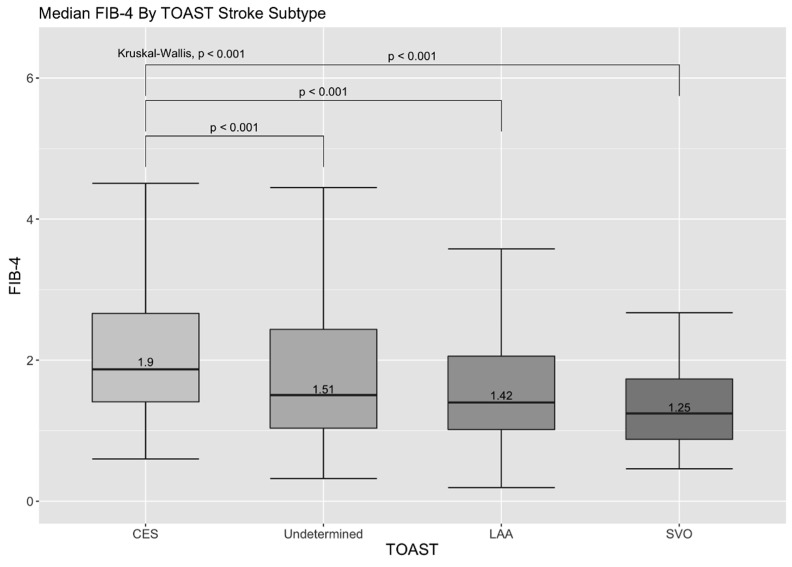
This figure is a boxplot depicting the median FIB-4 across the four different stroke mechanisms according to the TOAST classification, namely Cardioembolic stroke, Stroke of Undetermined Etiology, Large Artery Atherosclerosis and Small Vessel Occlusion. Error bars represent the 25th and 75th quartile of FIB-4 for each TOAST mechanism. Mann–Whitney U test determined that there was a significant difference between FIB-4 between Cardioembolic stroke and each of the four TOAST stroke subtypes groups at *p* < 0.001. The Kruskal–Wallis test, which determined if there was overall a statistically significant difference between any one of the four groups, was also statistically significant at *p* < 0.001.

**Table 1 jcm-12-00356-t001:** Baseline Characteristics of patients, stratified according to FIB-4 cut-offs.

	Total	FIB-4 < 1.3	1.3 ≤ FIB-4 ≤ 2.67	FIB-4 > 2.67	*p*
Total Number of Patients	887	342	384	161	
Age (years, median (IQR))	67 (57, 77)	56 (48, 63)	71 (63, 77.25)	81 (72, 87)	<0.001
Male (*n*, %)	528 (59.5)	223 (65.2)	224 (58.3)	81 (50.3)	0.005
FIB-4 (median (IQR))	1.53 (1.05, 2.27)	0.95 (0.75, 1.11)	1.77 (1.51, 2.12)	3.54 (3.05, 4.48)	<0.001
BMI (kg/m^2^, median (IQR))	24.46 (22.48, 27.36)	25.02 (23.16, 27.96)	24.24 (22.11, 27.36)	23.88 (20.78, 26.65)	<0.001
Race (*n*, %)					0.001
Chinese	535/786 (68.1)	185/305 (60.7)	236/333 (70.9)	114/148 (77.0)	
Indian	163/786 (20.7)	70/305 (23.0)	69/333 (20.7)	24/148 (16.2)	
Malay	48/786 (6.1)	24/305 (7.9)	20/333 (6.0)	4/148 (2.7)	
Others	40/786 (5.1)	26/305 (8.5)	8/333 (2.4)	6/148 (4.1)	
Hypertension (*n*, %)	128/887 (14.4)	70/342 (20.5)	50/384 (13.0)	8/161 (5.0)	<0.001
Hyperlipidemia (*n*, %)	469/887 (52.9)	162/342 (47.4)	226/384 (58.9)	81/161 (50.3)	0.006
Diabetes Mellitus (*n*, %)	592/887 (66.7)	194/342 (56.7)	277/384 (72.1)	121/161 (75.2)	<0.001
Atrial Fibrillation (*n*, %)	277/549 (50.5)	109/217 (50.2)	130/239 (54.4)	38/93 (40.9)	0.086
Stroke Parameters					
Admitting NIHSS (median (IQR))	15.5 (8, 21)	12 (7, 19)	15 (8, 21)	21 (16, 24)	<0.001
Moderate to severe stroke (NIHSS ≥ 10) (*n*/total, %)	619/880 (70.3)	208/340 (61.2)	269/380 (70.8)	142/160 (88.8)	<0.001
Admitting SBP (mmHg, median (IQR))	152 (136, 168)	151 (135, 166)	151 (135, 168)	160 (141, 171.5)	0.014
Admitting DBP (mmHg, median (IQR))	82 (72, 91)	83 (74.75, 94)	80 (71, 90)	81 (72.5, 91.5)	0.134
Large vessel occlusion (*n*/total, %)	543/811 (67.0)	191/320 (59.7)	230/345 (66.7)	122/146 (83.6)	<0.001
Onset-to-treatment time (min, median (IQR))	158 (118.75, 206.25)	153 (118, 198)	163 (123, 207)	156.5 (113.25, 209.25)	0.309
TOAST (*n*/total, %)					<0.001
Large-artery atherosclerosis	291/864 (33.7)	126/331 (38.1)	116/377 (30.8)	49/156 (31.4)	
Cardioembolism	218/864 (25.2)	44/331 (13.3)	117/377 (31.0)	57/156 (36.5)	
Small-vessel occlusion	137/864 (15.9)	72/331 (21.8)	58/377 (15.4)	7/156 (4.5)	
Stroke of other determined etiology	11/864 (1.3)	6/331 (1.8)	4/377 (1.1)	1/156 (0.6)	
Stroke of undetermined etiology	207/864 (24.0)	83/331 (25.1)	82/377 (21.8)	42/156 (26.9)	

Values are median (IQR) for numerical variables and *n*/total (%) for categorical variables. Abbreviations: IQR Interquartile range, FIB-4 Fibrosis-4, BMI body mass index, HbA1c hemoglobin A1c, LDL-C low-density lipoprotein cholesterol, HDL-C high-density lipoprotein cholesterol, AST aspartate transaminase, ALT alanine transaminase, NIHSS National Institutes of Health Stroke Scale, SBP systolic blood pressure, DBP diastolic blood pressure, TOAST The Trial of Org 10172 in Acute Stroke Treatment, mRS modified Rankin Scale.

**Table 2 jcm-12-00356-t002:** Predictors for Symptomatic Intracranial Hemorrhage using Multivariable Logistic Regression.

	SICH			Mortality			Functional Independence		
	Adj OR	95% CI	*p*-Value	Adj OR	95% CI	*p*-Value	Adj OR	95% CI	*p*-Value
FIB-4 > 2.67	4.089	1.035–16.160	0.045	3.052	1.283–7.262	0.012	0.998	0.531–1.876	0.995
Age	0.993	0.950 -1.039	0.771	1.033	1.002–1.065	0.037	1.050	1.029–1.073	<0.001
Gender	0.744	0.288–1.924	0.542	1.421	0.733–2.756	0.299	0.912	0.578–1.438	0.691
Admitting NIHSS	1.038	0.962–1.120	0.335	1.148	1.084–1.215	<0.001	1.093	1.056–1.132	<0.001
LVO	3.480	0.442–27.378	0.236	1.205	0.359–4.040	0.763	1.911	0.990–3.691	0.058
Hypertension	1.090	0.292–4.063	0.898	0.708	0.318–1.579	0.399	0.920	0.536–1.579	0.762
Hyperlipidemia	3.993	1.222–13.044	0.022	1.322	0.640–2.731	0.450	0.847	0.516–1.392	0.514
Diabetes mellitus	0.655	0.248–1.731	0.394	2.697	1.413–5.146	0.003	1.980	1.248–3.142	0.004
Body Mass Index	1.070	0.953–1.202	0.252	1.087	1.001–1.181	0.046	1.007	0.953–1.064	0.802
Atrial fibrillation	0.493	0.080–3.044	0.446	0.781	0.175–3.484	0.746	2.158	0.579–8.047	0.252
TOAST									
Cardioembolism	2.936	0.477–18.082	0.246	1.070	0.250–4.571	0.927	0.508	0.142–1.822	0.299
Small-vessel occlusion	1.982	0.114–34.392	0.638	0.597	0.086–4.127	0.601	0.810	0.344–1.907	0.631
Stroke of other determined etiology	37.184	3.391–407.768	0.003	12.405	1.192–129.095	0.035	1.517	0.540–4.261	0.687
Stroke of undetermined etiology	1.125	0.318–3.980	0.855	0.516	0.223–1.195	0.123	0.712	0.413–1.227	0.221

Abbreviations: Adj OR Adjusted Odds Ratio, 95% CI 95% confidence interval, SICH symptomatic intracranial haemorrhage, FIB-4 Fibrosis-4, NIHSS National Institutes of Health Stroke Scale, LVO large vessel occlusion, BMI body mass index, TOAST The Trial of Org 10172 in Acute Stroke Treatment.

## Data Availability

The data presented in this study are available on request from the corresponding author. The data are not publicly available due to ethical restrictions.

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
