# Peer review of "Risk of Liver Fibrosis Is Associated with More Severe Strokes, Increased Complications with Thrombolysis, and Mortality"

_jcm, 2023, doi:10.3390/jcm12010356_

Round 1

Reviewer 1 Report

The authors evaluated whether hepatic fibrosis causes an increased risk of intracranial hemorrhage, mortality and disability in patients with ischemic stroke undergoing thrombolysis. Liver function was assessed by the FIB-4 index

The main limitation of this study is its retrospective and single center character with consequent effects on the data presented. In particular, the Authors point out that patients with FIB-4 index >2.67 have very different baseline characteristics compared to patients with FIB-4 index <1.3 and although in the subsequent regression analysis the authors correct for a series of cofactors, the final result at which they arrive can still be affected by the differences at the baseline and therefore not credible. To minimize this limit, this reviewer suggests applying propensity score matching to populations with FIB-4 index >2.67 and FIB-4 index <1.3 for all those factors that show significant differences at baseline between these two populations. This analysis may or may not reinforce the effect of liver function on outcomes, net of the effects of other covariates. In each case the credibility of the data would be strengthened.

Author Response

Please see attachment, thank you!

Reviewer 2 Report

Dear authors

I read the manuscript entitled” Liver Fibrosis is associated with More Severe Strokes, Increased Complications with Thrombolysis, and Mortality “. In this original article, the authors investigated the association of Fibrosis (FIB)-4 index with outcomes after acute ischemic stroke. Hence, I have provided some comments as follows:

Major Comments

1-    How was fibrosis confirmed in patients? For example, did you do Ultrasound elastography or other techniques?

2-    How did the authors categorize the patients regarding the stages of the disease? For example, METAVIR scoring system.

3-    The authors sufficed to say, “The FIB-4 score 30 may be useful for prognostication after IVT”. How did you prove your claim? Did you depict ROC curve to present sensitivity and specificity of FIB-4 test?

4-    The aim at the end of introduction and in the abstract are not matched and seem confusing. For instance: the association between the FIB-4 index and the outcomes of AIS after IV tPA therapy (line 61) but the association of FIB-4 with outcomes after AIS (line 19)

5-     In addition, the authors used some words in the main text that do not exist in the abstract. For example, IV tPA therapy (line 62) and intravenous thrombolysis (IVT)

6-    Which declaration was used for patients? Helsinki?

7-    Which P-value was considered statistically significant?

8-    When we consider NAFLD, we can also use Gama-Glutamyl Transferase (GGT) with higher sensitivity. Why was this enzyme not assayed? In addition, other inflammatory parameters such as CRP? METAVIR scoring system could help discriminate inflammation (activity) and damage (fibrosis) in the liver.

9-    In figure 1, the authors categorized the patients into four groups but did not identify them in the methods. They were classified on what basis? Furthermore, they used Kruskal-Wallis test, but this test has not been mentioned in statistical analysis.

Minor Comments

1-    The paper needs to be edited by a native speaker. For example, in line 286, FIB-4 is a novel marker that could prognosticate IV tPA use in AIS.

2-    The paper is full of abbreviations. If some of them are not needed, use full words of them to help readers focus on the main concepts.

Reviewer 3 Report

It is a retrospective study at a comprehensive stroke centre. The study “Liver Fibrosis is associated with More Severe Strokes, Increased Complications with Thrombolysis, and Mortality” is to explore that the association between FIB-4 index and clinical outcomes among patients treated with IV tPA for acute ischemic stroke (AIS). The study disclosed that patients with FIB-4>2.67 had higher risk of cardioembolic stroke subtype and more moderate to severe strokes, were associated with an increased risk of SICH after thrombolysis as well as increased mortality post-AIS and IVT compared to patients with FIB-4<1.3. The results clearly presented the severity of liver fibrosis is associated with an increased risk of SICH which might be related to the higher mortality and morbidity. It is a valuable finding to emphasized the importance of evaluation of liver function before undertaking thrombolysis for patients with acute ischemic stroke to prevent against the complication of post-thrombolytic ICH. However, one minor question for clarification, were the patients recruited by the NINDS criteria only or both NINDS criteria and ECASS-III criteria in the study?

Author Response

Response to Reviewer 3:

Reviewer: It is a retrospective study at a comprehensive stroke centre. The study “Liver Fibrosis is associated with More Severe Strokes, Increased Complications with Thrombolysis, and Mortality” is to explore that the association between FIB-4 index and clinical outcomes among patients treated with IV tPA for acute ischemic stroke (AIS). The study disclosed that patients with FIB-4>2.67 had higher risk of cardioembolic stroke subtype and more moderate to severe strokes, were associated with an increased risk of SICH after thrombolysis as well as increased mortality post-AIS and IVT compared to patients with FIB-4<1.3. The results clearly presented the severity of liver fibrosis is associated with an increased risk of SICH which might be related to the higher mortality and morbidity. It is a valuable finding to emphasized the importance of evaluation of liver function before undertaking thrombolysis for patients with acute ischemic stroke to prevent against the complication of post-thrombolytic ICH. However, one minor question for clarification, were the patients recruited by the NINDS criteria only or both NINDS criteria and ECASS-III criteria in the study?

Response: Thank you so much for your detailed comment and your comments. We greatly appreciate your kind feedback. In response to your clarification, both the NINDS criteria 1 and ECASS-III2 were used to evaluate eligibility for rTPA in this study. The NINDS rTPA stroke study evaluated neurological outcomes at 3 months for patients given rtPA within 3h of stroke onset. The ECASS III study assessed the efficacy of IV thrombolysis within 3h to 4.5h of stroke onset. Both these randomized trials concluded that giving IV thrombolysis within 3 or 4.5h of stroke onset let to favourable outcomes, even in the unadjusted analysis of the primary end point and the results remained significant after adjustment for all prognostic baseline characteristics. Since our study involved patients treated with IV rTPA within 4.5h of stroke onset, it was reasonable for us to use both the NINDS and ECASS-III criteria.

References:

  1. Tissue Plasminogen Activator for Acute Ischemic Stroke. New England Journal of Medicine. 1995;333(24):1581-1588.
  2. Hacke W, Kaste M, Bluhmki E, et al. Thrombolysis with Alteplase 3 to 4.5 Hours after Acute Ischemic Stroke. New England Journal of Medicine. 2008;359(13):1317-1329.

Round 2

Reviewer 1 Report

The Authors tried to perform propensity analysis but they verified that the sample size does not allow that approach and therefore they decide to not include propensity analysis in the manuscript. However, They have better described the limitations of the study in the "Discussion". Since the propensity score is not used, please delete the following sentence from the discussion: "...... and this association held in the propensity-matched sample" (lines 203-204 of the Discussion)  

Author Response

Response to Reviewer 1

Reviewer 1: The Authors tried to perform propensity analysis but they verified that the sample size does not allow that approach and therefore they decide to not include propensity analysis in the manuscript. However, They have better described the limitations of the study in the "Discussion". Since the propensity score is not used, please delete the following sentence from the discussion: "...... and this association held in the propensity-matched sample" (lines 203-204 of the Discussion) 

Response: Thank you so much for your kind response, acknowledging our efforts and thoroughly checking the paper. We have removed the line as per your request and apologise for the error made.

Reviewer 2 Report

I appreciate your hard efforts to revise the manuscript as I wanted. You kindly edited the aim of the study, Kruskal-Wallis, Helsinki declaration, limitations etc., but you only added ROC curve analysis to the authors' response not the text. In this letter, the authors have mentioned figure 1 A and B, but in the revised manuscript, figure 1 points to a boxplot depicting the median FIB-4 across the four different stroke mechanisms according to the TOAST classification.  The figures should be reconsidered.  I think the final version has not been uploaded correctly. Take care of the main text to cite the figures correctly. Except for this comment, other comments have been applied point-by-point. Good luck
